# Evaluation of outcome reporting in clinical trials of physiotherapy in bronchiectasis: The first stage of core outcome set development

**Hayat Hamzeh**[1]*, **Sally Spencer**[1,2,3], **Carol Kelly**[1,2], **Samantha Pilsworth**[1]

**1** Faculty of Health, Social Care & Medicine, Edge Hill University, Ormskirk, United Kingdom, **2** Cardio-Respiratory Research Centre, Edge Hill University, Ormskirk, Lancashire, United Kingdom, **3** Health Research Institute, Edge Hill University, Ormskirk, Lancashire, United Kingdom

* hamzehh@edgehill.ac.uk

## Abstract

### Introduction

The aim of this study is to explore outcomes currently reported in physiotherapy trials for bronchiectasis and investigate the level of consistency in outcome reporting. This mapping of outcomes will be used to inform the development of a core outcome set (COS) for physiotherapy research in bronchiectasis. Outcomes reported in randomised clinical trials (RCTs) and RCT protocols were reviewed and evaluated. We included trials with physiotherapy as the main intervention, including pulmonary rehabilitation, exercise prescription, airway clearance, positive expiratory pressure devices, breathing training, self-management plans, and home exercise program. Medline, CINAHL, Scopus, Cochrane Central Register of Controlled Trials (CENTRAL), and the physiotherapy evidence database (PEDro) were searched from inception using a prespecified search strategy. Records including adult patients with bronchiectasis were included. Outcomes were listed verbatim and categorised into domains based on a pre-specified system, frequency of reporting and sources of variation were inspected.

### Results

Of 2158 abstracts screened, 37 trials (1202 participants) and 17 trial protocols were identified. Eighteen different physiotherapy techniques were investigated. A total of 331 outcomes were reported. No single outcome was reported by all trials. The most reported outcomes were lung function (27 trials, 50%), health related quality of life (26 trials, 48.1%), and dyspnoea (18 trials, 33.3%). A list of 104 unique outcomes covering 23 domains was created. Trials focus on physiological outcomes, mainly those related to respiratory system functions. Outcomes related to functioning and life impact are often neglected.

### Conclusion

Outcome reporting in physiotherapy research for bronchiectasis was found to be inconsistent in terms of choosing and defining outcomes. Developing a core outcome set in this area

**Data Availability Statement:** Data are available from the Edge Hill University Figshare repository (DOI: 10.25416/edgehill.22179488).

**Funding:** This doctoral research is supported by a Graduate Teaching Assistant studentship from Edge Hill University, UK. The funders had no role in study design, data collection and analysis, decision to publish, or preparation of the manuscript.

**Competing interests:** The authors have declared that no competing interests exist.

of research is needed to facilitate aggregation of future trial results in systematic reviews that will in turn inform the strength of evidence for the effectiveness of physiotherapy. Outcome choice should include all stakeholders, including patients.

## Trial registration

This study is registered in the PROSPERO registry under the number CRD42021266247.

## Introduction

Bronchiectasis is a chronic respiratory disease characterised by widening and thickening of the airways, leading to accumulation of secretions and recurrent infections [1, 2]. The prevalence of bronchiectasis has increased globally in recent years [3], causing a substantial economic burden [4]. In the United Kingdom (UK), bronchiectasis-related mortality is more than twice that of the general population [5], while 5-years mortality rate is 12.4% in European population [6].

Physiotherapy is recognised as a core element of bronchiectasis care [7–9], however, it currently lacks high quality evidence of its effectiveness [10, 11]. This is partly attributable to difficulties in aggregating data from clinical trials in systematic reviews, due to inconsistent outcome reporting and variation of measurement instruments [12–17]. Consequently, using COMET (Core Outcome Measurement in Effectiveness Trials) methodology to standardise outcome reporting is important for improving the design of future trials [18].

The Core Outcome Set for PHysiotherapy trials in adults with BronchiEctasis project (COS-PHyBE) aims to develop a core outcome set for physiotherapy research in bronchiectasis [19]. The first phase is a review of literature, with the aim of capturing all important outcomes that are currently reported and using them to create an initial long list of outcomes for the subsequent consensus exercise.

The COMET initiative encourages researchers to review literature as an initial step to inform the Delphi consensus process [20]. By 2019, 93 outcome reporting reviews were completed, of which 80% were described as systematic reviews [21]. Previous outcome reporting reviews have highlighted the inconsistency of outcome reporting in many healthcare areas [21]. These inconsistencies manifested as variation of outcomes used, differences in outcome definitions, time points, and measurement methods [22–26]. Available systematic reviews of physiotherapy effectiveness in bronchiectasis have highlighted the inconsistency in outcome reporting among trials, which limited the aggregation of results and lead to inconclusive recommendations [12, 13 16, 17, 27, 28]. However, the variation in outcome reporting was not previously examined in this area.

The aim of this study was to identify and evaluate current clinical outcome measurement in bronchiectasis trials which investigated physiotherapy interventions. This mapping of outcomes will be used to inform the development of a COS for physiotherapy research in bronchiectasis. More specifically, the two main objectives were (1) creating a synthesised long-list of outcomes reported in literature and (2) assessing the variation in outcome reporting among relevant trials. Evaluating the level of inconsistency among available trials will determine the need for a COS in this area. This evaluation will be in terms of the number, variability, and definition of outcomes.

A search of Medline, CENTRAL, and PEDro databases and international prospective register of systematic reviews (PROSPERO) identified no similar published or registered reviews.

# Method

## Registration

The protocol was developed and is registered in the PROSPERO registry under the number CRD42021266247.

## Eligibility criteria

Studies of the effectiveness of physiotherapy for bronchiectasis were included. Studies which included patients with multiple respiratory conditions were excluded as the focus is to find outcomes used specifically for bronchiectasis, not the ones which may be useful across respiratory conditions. Full publications, pilot studies, and protocols of Randomised Controlled Trials (RCTs), controlled clinical trials, quasi-randomised studies and crossover studies were included.

Registered and published study protocols were included as they provide comprehensive discussion of outcomes and measurement methods. They also reflect recent research being conducted but not yet published. Studies published only as conference abstracts were excluded, as they did not provide adequate data regarding outcome measurement because of the limited word count. Full Inclusion and exclusion criteria for this review are summarised in Table 1.

## Information sources and search strategy

Medline, CINAHL, Scopus, Cochrane Central Register of Controlled Trials (CENTRAL), and the physiotherapy evidence database (PEDro) were searched from inception to 01.09.2022 using a prespecified search strategy. All search results were verified by a second reviewer. The search was limited to English language only due to limited language translation resources. An example search strategy, used for Medline, is provided in S1 Appendix.

Relevant registered protocols of ongoing or unpublished studies were sought by searching the US National Institutes of Health Trials Register (ClinicalTrials.gov) and the International Clinical Trials Registry Platform (ICTRP). OpenGrey [29] and ProQuest dissertations and thesis [30] databases were searched for relevant grey literature, like theses, dissertations and

**Table 1. Study selection criteria.**

| PICOS criteria | Included | Excluded |
|---|---|---|
| Population | Patients with Bronchiectasis, confirmed radiologically on high resolution computed tomography, Adults 18 or above years of age As bronchiectasis is commonly accompanied by other diseases, participants with other comorbidities will be included | • Studies of mixed populations (e.g., participants with different respiratory diseases) • children <18 years of age |
| Interventions | Studies with physiotherapy as the main intervention, e.g.: rehabilitation, exercise prescription, airway clearance, positive expiratory pressure devices, breathing training, respiratory muscle training, self-management plans, home exercise program | Studies evaluating the use of adjuncts to physiotherapy (humidification or saline nebulization, ventilation, bronchodilators, nutrition, etc.) |
| Comparisons | Control, sham, no treatment, alternative physiotherapy intervention, usual care | None |
| Outcomes | Any outcomes | None |
| Type of study | • Randomised control trial, crossover trial. • Protocols of RCTs • Multiple publications of the same study will be included as one record. | • Articles which evaluate or describe the psychometric properties of a measurement instrument. • Studies of qualitative methods (e.g., semi-structured interview or primarily open-ended questions • Trials published in non-English language. |

conference abstracts. A manual search of references lists of relevant systematic reviews was conducted to identify any additional records.

## Selection process

All initial search results were exported to an Endnote software library (Clarivate Analytics). Duplicates were removed using the Endnote find duplicate function then revised manually. An Endnote web shared library was used to communicate selected and excluded records among two reviewers (HH and SP) who completed the selection process. The two reviewers independently screened titles and abstracts for eligibility against review selection criteria. Full texts of potentially eligible studies were then obtained and independently checked to confirm eligibility against inclusion and exclusion criteria. Selection decisions were discussed between the two reviewers. There were no unresolved disagreements that required referral to a third reviewer. The full process of selection is detailed in the Preferred Reporting Items for Systematic Reviews and Meta-Analyses (PRISMA) flowchart (Fig 1).

## Data collection process

All data were extracted and added into a Microsoft Excel spreadsheet. One reviewer (HH) extracted all data with 20% of the data verified by a second reviewer (SP). Study characteristics data included author, year, country, study design, sample size, interventions, and number of outcomes measured. For protocols, country, planned sample size, interventions and number of outcomes were extracted.

The following data were extracted for each outcome: outcome name, outcome definition (if available), and whether outcome is stated as primary or secondary (outcome used to calculate sample size was regarded as primary). When the study reported only measurement instruments, the corresponding outcome for that instrument was harvested from literature. Outcome data were sought from abstracts, methods, and results sections in published trials reports, and from dedicated outcomes and outcome measures sections in registers.

## Grouping and analysis of outcomes

**1. Creating unique outcome long list.** After outcomes and their definitions were extracted verbatim, outcomes were analysed according to COMET handbook guidelines [20]. Exact duplicate outcomes in wording and spelling were removed. Then, any outcomes with different spelling of the same words were regarded as duplicates and removed, e.g.: Dyspnea and dyspnoea. Composite instruments, i.e., instruments which measure multiple outcomes were classified under all relevant outcomes they cover. Outcomes with the same meaning and context are commonly described using different terminology and definitions, which leads to outcome lists being extremely long. Therefore, outcomes with similar definitions or measurement methods were regarded as having the same meaning. These outcomes were grouped, and two reviewers agreed upon a unique name for each outcome.

To facilitate understanding of the long list of outcomes, similar outcomes were grouped into themes, then classified into domains using the COMET outcome taxonomy [32]. Unlike other outcome classifications, this taxonomy provides a wide range of domains that covers all potential outcomes used in trials. It includes 38 outcome categories covering the core areas of death, physiological outcomes, life impact, resource use, and adverse effects. An outcome matrix was created according to the taxonomy to analyse the frequency of use of each outcome domain, by matching each trial with corresponding outcomes. This process produced the long list of unique outcomes. The long list was revised by two senior researchers to ensure adequate use of terms.

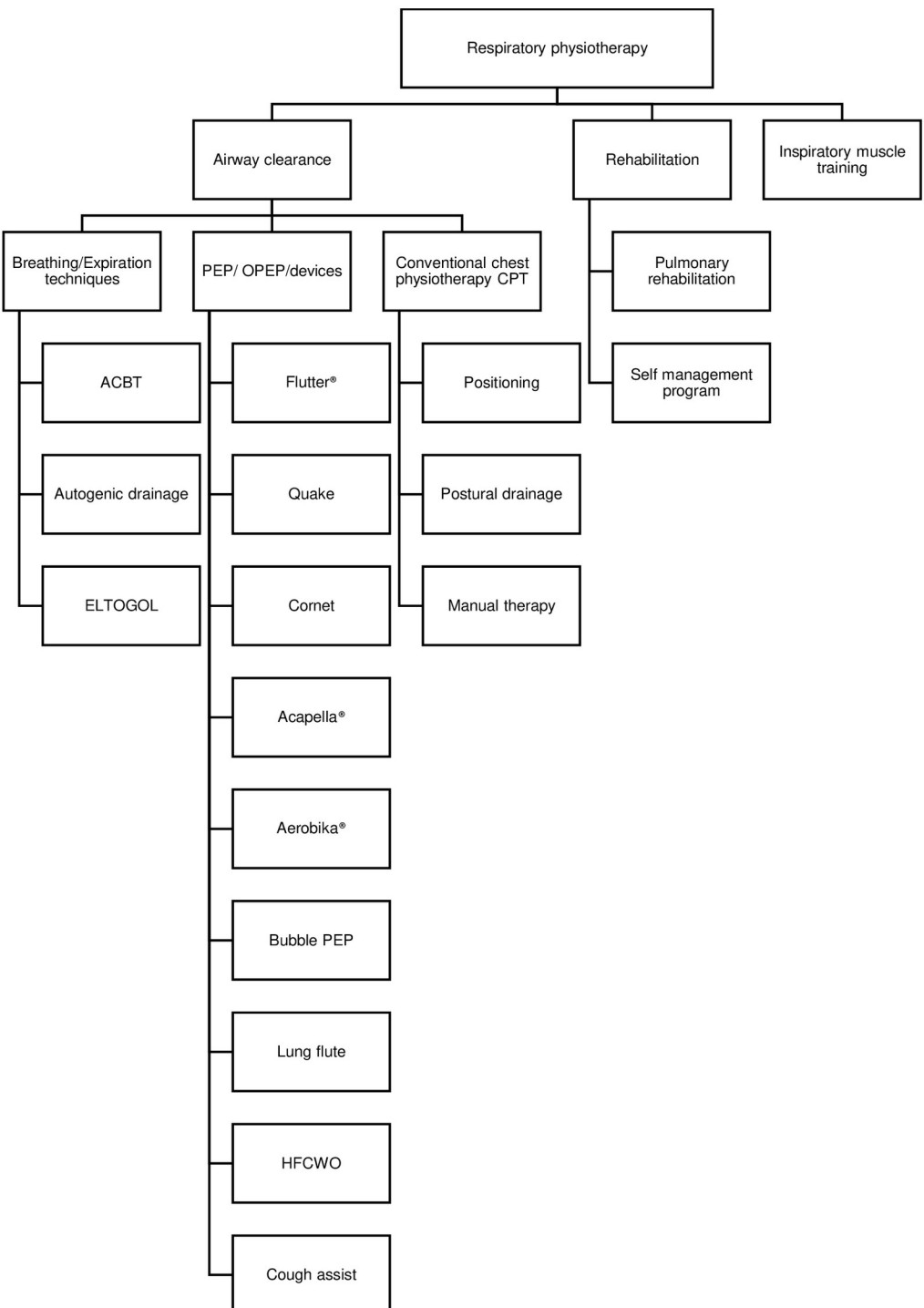

**Fig 1. Flow of studies through the review, PRISMA 2020 flow diagram** [31]**.** ACBT: active cycle breathing technique, CPT: chest physiotherapy, PEP: positive expiratory pressure, OPEP: oscillating positive expiratory pressure IMT: inspiratory muscle training, ELTGOL: slow expiration with the glottis opened in a lateral posture, HFWCO: high frequency chest wall oscillation.

**2. Evaluation of outcome reporting consistency.** Consistency of outcome reporting across studies was analysed following recommendations by Young and colleagues [33] based on the following: number of verbatim outcomes per study and across studies, number of unique outcomes per study and across studies, number of different terms to describe the same outcome across studies, and number of outcomes reported at each timepoint.

## Results

### Search results

A total of 2158 abstracts were identified from initial search; 1528 remained after removing duplicates. Screening titles and abstracts excluded 1388 records and 140 full texts were reviewed. Seventy-four reports of 37 studies and 17 study protocols were identified as eligible for inclusion (Fig 1). No additional eligible trials were identified from grey literature databases or review citation search.

### Study characteristics

Characteristics of trials included in the review is described in Table 2. Of these, 20 (55.6%) were crossover studies, 14 were RCTs, and two were controlled clinical trials (CCTs). The washout period in the crossover studies ranged between 12 hours and two weeks. A total of 1202 participants were recruited across the trials, ranging between 8 and 85 per trial. Only 5 (13.5%) trials recruited more than 50 participants, while 20 (54%) recruited 30 participants or less. The majority of trials recruited from a single site and only two (5.6%) had more than one recruitment site. Studies were published between 1999 and 2021 with 13 (35.1%) trials published in the last five years. The included studies represented 13 countries from different global regions. Nine (25%) trials were from the UK, followed by eight from Brazil then six from India and four from Australia.

### Interventions

Thirteen trials compared physiotherapy to control, sham, or placebo; while 23 trials compared two or more physiotherapy techniques using active comparator groups (Table 2). The effectiveness of multiple physiotherapy interventions covering both airway clearance and pulmonary rehabilitation were investigated, including a total of 18 different techniques (Fig 2). The two most investigated techniques were the active cycle of breathing technique (ACBT) and the Positive Expiratory Pressure devices (PEP). ACBT was the most investigated airway clearance technique in 10 trials (27.7%) which compared it to other physiotherapy techniques or control. The Flutter device was the most investigated PEP device in 10 trials (27.7%). The effectiveness of pulmonary rehabilitation was tested in seven RCTs, all used an 8-week program.

### Outcomes

A total of 331 outcomes were identified from the included trials and protocols. The number of outcomes reported per trial ranged from 1 to 29 with a median of 6. One trial reported a single outcome, 42.6% trials reported 5 or less outcomes, 33.3% reported between 6 and 10 outcomes, while 24.1% reported more than 10 different outcomes. Of the 331 outcomes, 51 outcomes (15.4%) were used only in one trial, while 91 (27.5%) were used in 5 trials or less (Fig 3, S3 Appendix). No single outcome was reported across all studies. The most reported outcomes were lung function (27 trials, 50%), health related quality of life (26 trials, 48.1%), and dyspnoea (18 trials, 33.3%). A total of 18 trials specified their primary and secondary outcomes. Most reported primary outcomes were sputum-related outcomes (10 trials) and exercise

**Table 2. Characteristics of included trials.**

| First author, Reference | Year | Country | Study design | No of sites | Sample Size | Interventions | Comparisons |
|---|---|---|---|---|---|---|---|
| Abdelhalim [34] | 2016 | Egypt | RCT | 1 | 30 | ACBT | conventional CPT (PD + DB + percussion) |
| Amit Vyas [35] | 2012 | India | Crossover | 1 | 35 | Quake PEP | RC-Cornet PEP |
| Cecins [36] | 1999 | Australia | Crossover | 1 | 19 | ACBT + head tilt | ACBT |
| Chalmers [37] | 2019 | UK | RCT | 1 | 27 | pulmonary rehabilitation | standard care |
| De Oliveira Antunes [38] | 2001 | Brazil | Crossover | 1 | 10 | Conventional CPT | Flutter VRP1 |
| De Souza Simoni [39] | 2019 | Brazil | Crossover | 1 | 40 | Flutter PEP | Manual therapy Control |
| Eaton [40] | 2007 | New Zealand | RCT | 1 | 36 | ACBT + PD | ACBT Flutter PEP |
| Figueiredo [41] | 2012 | Brazil | Crossover | 1 | 8 | Flutter PEP | Sham flutter |
| Guimarães [42] | 2012 | Brazil | Crossover | 1 | 10 | ELTGOL | Flutter PEP |
| Herrero-Cortina [43] | 2016 | Spain | Crossover | 1 | 31 | AD | ELTGOL Temp PEP |
| Jose [44, 45] | 2017 2021 | Brazil | RCT | 1 | 63 | Home based pulmonary rehabilitation | Control |
| Lavery [46] | 2011 | UK | RCT | 1 | 64 | Expert Patient Programme Self-management | Usual care |
| Lee [47, 48] | 2010 2014 | Australia | RCT | 3 | 85 | Exercise + ACT education | Control |
| Liaw [49] | 2011 | Taiwan | RCT | 1 | 26 | IMT | Control |
| Livnat [50] | 2021 | Israel | RCT | 1 | 51 | PEP Aerobica | AD |
| Mandal [51] | 2012 | UK | RCT | 1 | 30 | PR + CPT | CPT |
| Munoz [52] | 2018 | Spain | RCT | 2 | 44 | ELTGOL | Placebo |
| Murray [53] | 2009 | UK | Crossover | 1 | 20 | PEP acapella | no treatment |
| Naraparaju [54] | 2010 | India | Crossover | 1 | 30 | Acapella PEP | IMT |
| Newall [55] | 2005 | UK | RCT | 1 | 32 | PR + IMT | PR + Sham Control |
| Nicolini [56] | 2013 | Italy | RCT | 1 | 30 | HFCWO | Control |
| Oliveira dos Santos [57] | 2018 | Brazil | RCT protocol | 1 | 60 | pulmonary rehabilitation | Control |
| Ozalp [58] | 2019 | Turkey | RCT | 1 | 45 | IMT | Control |
| Patterson [59] | 2004 | UK | Crossover | 1 | 20 | test of incremental respiratory endurance (TIRE) | ACBT |
| Patterson [60] | 2005 | UK | Crossover | 1 | 20 | ACBT | Acapella |
| Patterson [61] | 2007 | UK | Crossover | 1 | 20 | Acapella | ACT |
| Ramos [62] | 2015 | Brazil | Crossover | 1 | 22 | coughing | PD PD + percussion PD + huffing |
| Santos [63] | 2020 | Australia | Crossover | 1 | 35 | bubble-PEP | ACBT control |
| Semwal [64] | 2015 | India | Crossover | 1 | 30 | AD | Acapella |
| Senthil [65] | 2015 | India | CCT | 1 | 30 | ACBT + Acapella | ACBT |
| Shabari [66] | 2011 | India | Crossover | 1 | 35 | Quake | RC cornet |
| Silva [67] | 2017 | Australia | Crossover | 1 | 40 | Flutter | Lung flute |
| Syed [68] | 2009 | India | Crossover | 1 | 35 | ACBT | CPT |
| Tambascio [69, 70] | 2011 2017 | Brazil | Crossover | 1 | 17 | Flutter | Sham flutter |
| Thompson [71] | 2002 | UK | Crossover | 1 | 17 | ACBT | Flutter |

(*Continued*)

**Table 2.** (Continued)

| First author, Reference | Year | Country | Study design | No of sites | Sample Size | Interventions | Comparisons |
|---|---|---|---|---|---|---|---|
| Tsang [72] | 2003 | Hong Kong | RCT | 1 | 15 | PD + breathing and cough training | Flutter + breathing and cough training<br>breathing and cough training |
| Üzmezoğlu [73] | 2018 | Turkey | CCT | 1 | 40 | ACBT | Flutter |

CCT: controlled clinical trial, ACBT: active cycle breathing technique, CPT: chest physiotherapy, PD: postural drainage, ACT: conventional airway clearance technique, PR: Pulmonary rehabilitation, PEP: positive expiratory pressure, IMT: inspiratory muscle training, ELTGOL: slow expiration with the glottis opened in a lateral posture, HFWCO: high frequency chest wall oscillation

capacity (6 trials). Noticeably, trials of airway clearance techniques did not measure outcomes related to exercise capacity and physical functioning, while pulmonary rehabilitation studies did not measure outcomes related to sputum production.

## Variation in outcome definitions

After removing exact duplications, 239 different outcomes remained. Some outcomes assessing the same context were described differently among trials, variations were mainly in wording of the description and not in the definition of the outcome. This wording variation occurred 49 times in total, ranging from 2 to 16 variations per outcome. For example, assessing pulmonary function, including static and dynamic Spirometric measurements, was described using 16 different terms (Box 1). Some trials did not provide clear outcome definitions, exacerbations were measured in five trials, only two of them used European Respiratory Society consensus as a definition while the others did not provide a definition. In many cases the description was restricted to the name of the instrument and not the outcome of interest. Some trials reported spirometry without defining which spirometry-measured outcomes they were reporting.

## Outcome categories and domains

After removing duplicates and choosing a unique name for outcomes with the same meaning, a final list of 104 outcomes was produced (S2 and S3 Appendices). Outcomes with similar context were grouped and classified into a total of 23 domains according to COMET taxonomy (Table 3). The most reported domain was respiratory outcomes, which was further classified into 6 different subdomains to facilitate understanding. Frequency of reporting each domain is represented in Fig 3.

## Discussion

This is the first review of literature to describe the variation in outcome reporting within the field of physiotherapy trials for bronchiectasis. The results demonstrated high variability in outcome reporting. This inconsistency was not limited to previously published trials but also extended to current ongoing trials, which predicts a continuous problem of research waste in the future [74].

Inclusion of only RCTs is common in similar outcome reporting reviews for COS development [22–26]. This is because the focus of a COS is improving future RCT designs and the quality of evidence they provide. Notwithstanding the potential methodological issues of crossover designs for physiotherapy studies, additional inclusion of crossover studies avoided missing any important outcomes from these studies. Other reviews included studies of all designs

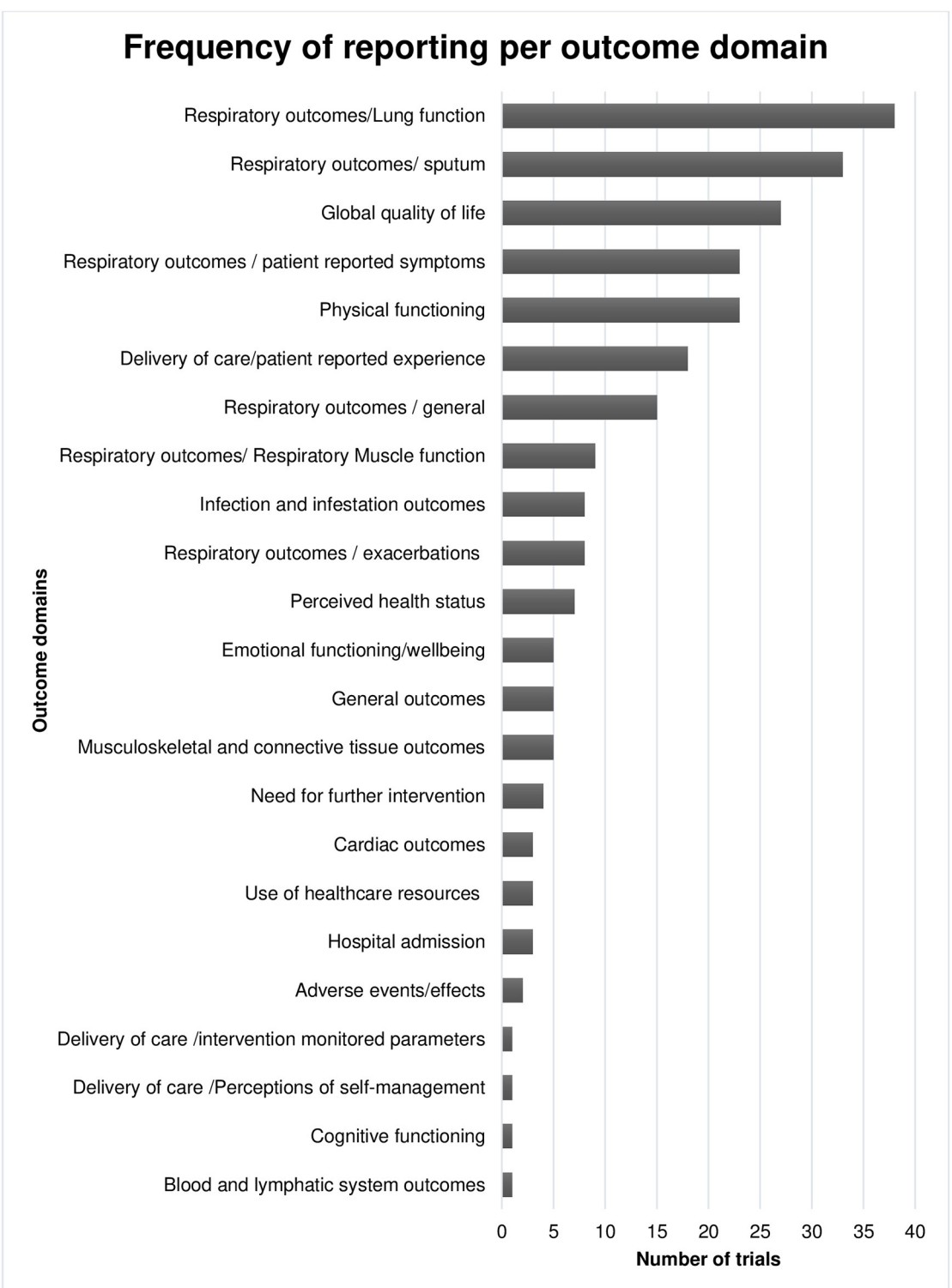

**Fig 2. Types of physiotherapy interventions used across included studies.**

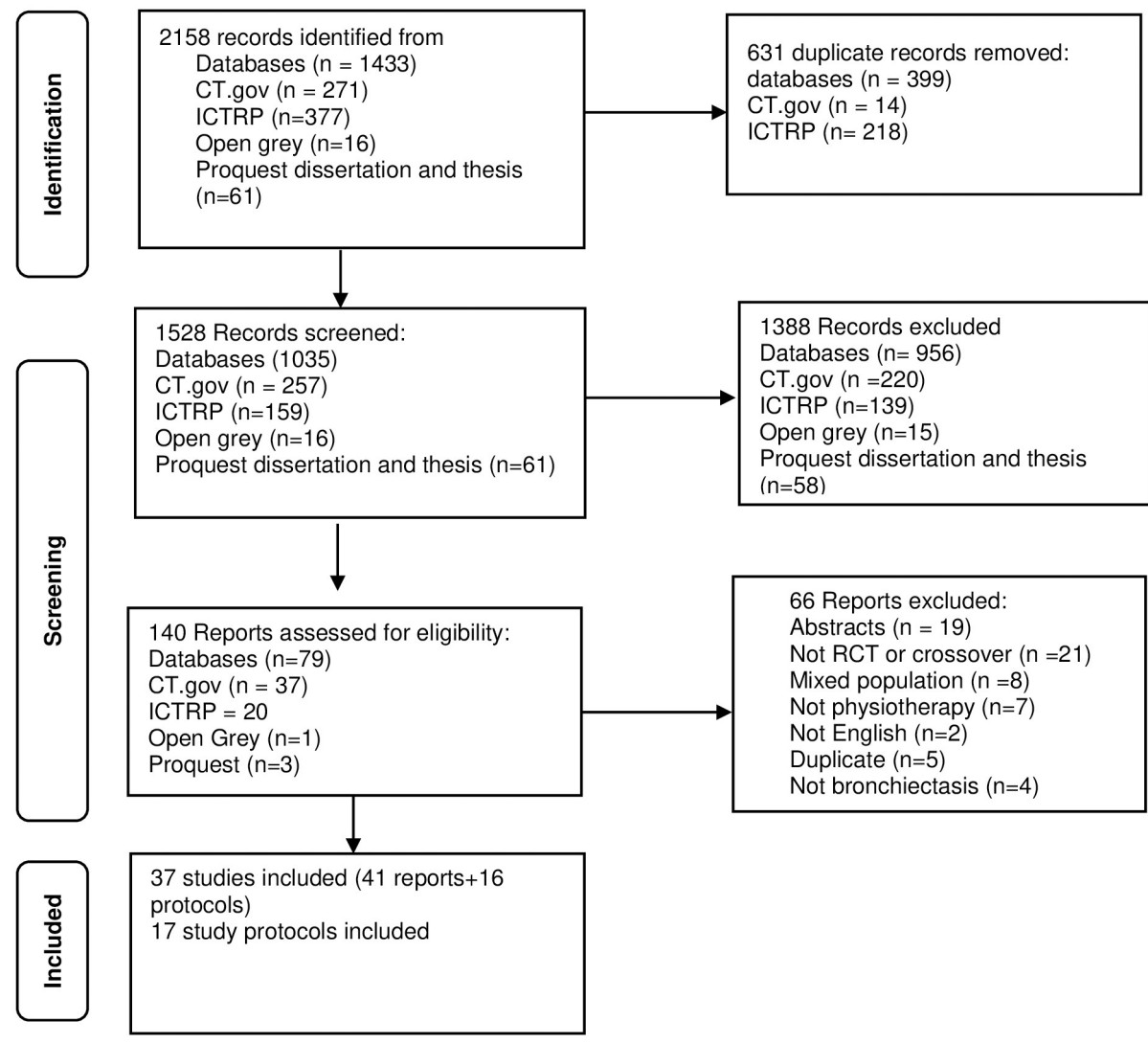

**Fig 3. Frequency of outcomes reported per trial.**

in order to collect a wider range of outcomes, this is of particular importance when there are limited RCTs available and the focus of the COS is to inform a wide range of study designs [75–77]. Some reviews also included qualitative studies in order to gather additional outcomes important to patients and the public [78]. Information from these stakeholder groups will be collated during the subsequent interviews stage and first round of the Delphi study.

## Inconsistency in outcome reporting

The main form of inconsistency was the variable selection and inclusion of outcomes across trials, as no single outcome was reported across all studies and 28% were used in less than 5 studies. Inconsistency was also manifested as incomplete or lack of definitions of outcomes and lack of primary outcomes. Similar inconsistencies were highlighted in multiple published reviews of outcome reporting in several healthcare areas, such as oncology, orthopaedics, neurology, surgery, nephrology and women's health [21].

Box 1. Wording used to describe the outcome 'pulmonary function'.

Lung function

Lung function testing

Pulmonary function

Pulmonary function index

Pulmonary function test

Pulmonary function test readings

Pulmonary function testing

Pulmonary function tests

Respiratory function

Respiratory function test

Spirometric lung function

Spirometric lung volumes

Spirometric measures of lung function

Spirometric parameters

Spirometry

Ventilatory function: post bronchodilator spirometric tests

One reason for inconsistency may be poor adherence to the Consolidated Standards of Reporting Trials (CONSORT) guidelines for trial reporting, which called for 'Completely defined pre-specified primary and secondary outcome measures, including how and when they were assessed' [79]. This was noted in cardiorespiratory physiotherapy trials where only a fifth of RCTs had specified their primary outcome, and this was linked to the poor overall quality of studies [80].

Lack of agreement on outcome selection is an evident problem in the bronchiectasis literature that affects interpretation of the evidence, as it limits data aggregation in systematic reviews. There is a lack of international consensus on selection of important outcomes both in general and in physiotherapy specific research. Physiotherapy important outcomes were defined by the American College of Chest Physicians (ACCP) as quality of life, mortality, hospital admission, and exacerbation rates [81]. The European Respiratory Society (ERS) guidelines encouraged researching the effectiveness of physiotherapy using outcomes of accessibility, patient preference and adherence [10]. The British Thoracic Society (BTS) recommended using clinically meaningful outcomes in bronchiectasis studies, but they did not name these outcomes in their published report [1].

Similarly, there is a noticeable gap between what guidelines committees deem as critical outcomes and what is being measured in trials. ACCP were unable to make recommendations regarding the effectiveness of airway clearance in bronchiectasis because the available trials did not target the outcomes that they considered meaningful [81]. ERS guidelines looked for hospitalisations; physical activity, adverse events, treatment burden, and fatigue and the available

**Table 3. Classification of outcomes, adapted from COMET taxonomy [32].**

| Core area | Outcome domain | Most reported outcome |
|---|---|---|
| Clinical/Physiological outcomes | Blood and lymphatic system outcomes | Blood cell count |
| | Cardiac outcomes | Heart rate |
| | General outcomes | Disease severity |
| | Infection and infestation outcomes | Blood inflammation markers |
| | Musculoskeletal and connective tissue outcomes | Muscle strength |
| | Respiratory, thoracic and mediastinal outcomes | Oxygen saturation (SPO2) |
| | Respiratory, thoracic and mediastinal outcomes: lung function | Pulmonary function |
| | Respiratory, thoracic and mediastinal outcomes: sputum | Sputum weight |
| | Respiratory, thoracic and mediastinal outcomes: patient reported symptoms | Breathlessness |
| | Respiratory, thoracic and mediastinal outcomes: respiratory muscle function | Maximal Inspiratory Pressure (PImax) |
| | Respiratory, thoracic and mediastinal outcomes: exacerbations | Exacerbation frequency |
| Life impact | Physical functioning | Six minute walk distance |
| | Emotional functioning/wellbeing | Anxiety and depression |
| | Cognitive functioning | Cognitive loss |
| | Global quality of life | Health-related quality of life (HRQOL) |
| | Perceived health status | General health status |
| Resource use | Delivery of care: patient reported experience | Patient preference |
| | Delivery of care: self-management ability | Self-rated ability to manage bronchiectasis |
| | Delivery of care: intervention monitored parameters | Number of sets performed during session |
| | Hospital admission | Number of urgent hospital admissions |
| | Use of healthcare resources | Number of urgent/unplanned outpatient visits |
| | Need for further intervention | Antibiotics use |
| Adverse events/effects | Adverse events/effects | Side effects |

trials did not measure these outcomes [82]. Having a group of outcomes that are acceptable by both trialists and guidelines committees will facilitate creating evidence based clinical guidelines.

Predefining outcomes is an essential step in systematic reviews to reduce selective outcome reporting bias. Accordingly, reviewers should define outcome domains and outcome measures of interest to be included in the analysis [83]. As the most appropriate outcome in bronchiectasis has not yet been defined, systematic reviews are currently using various outcomes depending mainly on the reviewers' own choice. Developing a COS will help reviewers choose common important outcomes, alongside any additional outcomes specific for the review topic. It will also encourage trialists to consider these outcomes in future research. In a systematic review which defined exacerbation frequency as the main outcome, physiotherapy trials were not included in the results or meta-analysis because they did not measure exacerbations [27]. Another systematic review of the effects of positive expiratory pressure defined their outcomes as quality of life, rate of exacerbations, and risk of hospitalisation [84]. But they found that the trials only measured quality of life and used different measurement instruments. Therefore, they were unable to perform meta-analyses using their pre-defined outcomes. The use of some outcomes like exacerbations and sputum is still debated among experts which causes more divergence in research trials [85].

The choice of primary outcomes varied slightly according to the nature of the interventions. Airway clearance trials did not measure exercise capacity and physical functioning, while pulmonary rehabilitation studies did not measure sputum-related outcomes. Similarly, available systematic reviews of pulmonary rehabilitation determined exercise capacity as primary

outcome [17, 28], while airway clearance reviews were more interested in exacerbations, quality of life, and hospitalisation as primary outcomes [16, 84]. Consequently, subdividing the COS according to treatment will be considered during the consensus phase of this project.

Outcome reporting is not consistent even among studies of similar interventions. For example, the ACBT was investigated in four trials, including two randomised crossover studies [36, 68] and two randomised parallel group studies [34, 40], but sputum was the only common outcome among them. Using sputum as an outcome is controversial, as some authors argue it is not accurate because patients may swallow or be unwilling to expectorate secretions [86]. Also, patients are unsure whether sputum amount expectorated reflects better prognosis [85].

## Missing and under-reported outcomes

The results reflect a great focus on physiological outcomes, mainly those related to respiratory system functions. Although the main focus of physiotherapy treatment is improving human functioning [87, 88], these outcomes are poorly reported when compared to clinical and physiological outcomes. Death, survival, and mortality are important outcomes that are usually reported in effectiveness trials [32]. This domain was not reported in trials, despite the considerable mortality rates reported in bronchiectasis [6]. Physiotherapy potentially affects mortality as it may prevent severe exacerbations [52]. Use of healthcare resources, like hospital and ICU admissions were seldom reported, despite recurrent occurrence while living with this chronic disease. Adverse effects were reported only twice, although some physiotherapy techniques like postural drainage and manual techniques are known to have side effects [89]. Measurement of adverse effects is recommended as it provides a balanced perspective regarding the risks and benefits of interventions [90].

## Limitations

Including only English language articles may have limited the scope of outcomes collected in this review. Therefore, all efforts will be taken to encourage international participation in the Delphi phase of COS development project in order to include all important outcomes.

## Conclusions

Outcome reporting in research on physiotherapy for bronchiectasis was found to be inconsistent in terms of choosing and defining outcomes. Developing a COS in this area of research is needed to facilitate aggregation of future trial results in systematic reviews that will in turn inform the strength of evidence for the effectiveness of physiotherapy.

This review represents the important initial steps in the development of a COS for physiotherapy research in bronchiectasis, determining the list of outcomes currently used. The next step will be to investigate additional important outcomes identified by patients and clinicians, which will be added to this list. The long list of outcomes will then be used to develop an electronic Delphi prioritization exercise to reach consensus regarding the most important outcomes to measure in effectiveness studies of physiotherapy for bronchiectasis.

## Supporting information

**S1 Appendix. Search strategy *Ovid Medline*.**
(DOCX)

**S2 Appendix. The full list of outcomes.**
(DOCX)

**S3 Appendix. Frequency of reporting per outcome, calculated by number of trials and protocol that reported outcome.**
(DOCX)

**S1 Checklist. Preferred Reporting Items for Systematic reviews and Meta-Analyses extension for Scoping Reviews (PRISMA-ScR) checklist.**
(PDF)

## Author Contributions

**Conceptualization:** Hayat Hamzeh, Sally Spencer, Carol Kelly.

**Data curation:** Hayat Hamzeh, Samantha Pilsworth.

**Formal analysis:** Hayat Hamzeh, Samantha Pilsworth.

**Methodology:** Hayat Hamzeh, Sally Spencer.

**Project administration:** Hayat Hamzeh.

**Supervision:** Sally Spencer, Carol Kelly.

**Validation:** Samantha Pilsworth.

**Writing – original draft:** Hayat Hamzeh.

**Writing – review & editing:** Sally Spencer, Carol Kelly, Samantha Pilsworth.

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
