## [Decision Letter · Decision Letter 0]

20 Dec 2022

PONE-D-22-28058Evaluation of outcome reporting in clinical trials of physiotherapy in bronchiectasis: The first stage of core outcome set developmentPLOS ONE

Dear Dr. Hamzeh,

Thank you for submitting your manuscript to PLOS ONE. After careful consideration, we feel that it has merit but does not fully meet PLOS ONE’s publication criteria as it currently stands. Therefore, we invite you to submit a revised version of the manuscript that addresses the points raised during the review process.

We look forward to receiving your revised manuscript.

Kind regards,

Brenda M. Morrow, PhD

Academic Editor

PLOS ONE

Journal Requirements:

Reviewers' comments:

Reviewer's Responses to Questions

**Comments to the Author**

1. Is the manuscript technically sound, and do the data support the conclusions?

Reviewer #1: Yes

Reviewer #2: Yes

2. Has the statistical analysis been performed appropriately and rigorously? 

Reviewer #1: N/A

Reviewer #2: N/A

3. Have the authors made all data underlying the findings in their manuscript fully available?

Reviewer #1: Yes

Reviewer #2: Yes

4. Is the manuscript presented in an intelligible fashion and written in standard English?

Reviewer #1: No

Reviewer #2: Yes

5. Review Comments to the Author

Reviewer #1: This is an interesting and important study establishing what outcomes are currently reported for clinical trials of physiotherapy for people with bronchiectasis. Some points of clarification are required, and suggestions are detailed below.

Abstract

Page 2 – define COS in full on line 32.

It is not clear what the numbers after lung function, health related quality of life etc mean. Presumably, these are n=, but this perhaps needs to be made clearer to the reader. This also applies within the results section of the manuscript.

Was a pre-specified system for classifying into domains used?

page 4, line 71 - COS-PHyBE needs to be defined for your readers.

Page 5, line 88 – needs rephrasing “More specifically, the two main objectives are creating a synthesised long list 89 of outcomes and assessing the variation in outcome reporting.’

Under results and search results, the numbers quoted don’t completely add up – if 1528 remained, then a further 1388 were removed, this leaves 140 full texts rather than 141?

Page 9, line 193 – incomplete sentence ‘While 23 193 compared one form of physiotherapy to another by including an active comparator group 194 (table 2).’

On page 9, flutter device is described as a technique, but the actual technique is oscillating PEP or oscillatory PEP, the flutter device is just what is used to deliver this technique. This needs to be corrected to accurately reflect the type of technique rather than the device chosen to deliver it.

Not sure what is meant by the three categories of airway clearance techniques? Can the authors elaborate on this? I can see it is later stated in the figure, but it is still not clear what the difference is between airway clearance techniques PEP/OPEP etc and positioning. Wondering if the authors are referring to breathing techniques when they mean airway clearance techniques, following the ERS definition (from 2017). If so, this needs to be made clearer in the text.

In the paragraph of Interventions under results, there is a lack of references when stating 13 trials compared physiotherapy to sham, etc. It would be useful to have links to the relevant studies in this paragraph.

Some alterations to grammar are required within the manuscript to improve clarity and readability.

On page 10, it is not clear how the decision to choose a unique name presumably based on the COMET taxomony for outcomes with the same meaning was undertaken. Was this done by one author or a consensus?

In Table 2, it is not clear how the decision was made for what is the intervention versus the comparison? It implies that the focus of the study was on the intervention, but for some included studies, they compared 3 techniques, with no one technique the main focus. Can the authors provide some further clarity or consider reworking their description of interventions in this table? Also, there is inconsistency within the table, crossover and Crossover both appear – suggest fixing this.

In Table 3, some abbreviations are in ( ), others are not – suggest consistency throughout with careful attention to detail.

Figure 2 – what makes up CPT – as this has previously been PD as well, which would fit under positioning?

Not sure that Figure 3 adds much to the results, it is hard to understand the takeaway message from this?

Correct spelling of Aerobika and consider which devices are registered names and require ® at the end.

Appendix 2 – consider putting SpO2 in ( ) like the rest of abbreviations. Same for QALYs. Also needs to be adjusted in Appendix 3.

LCI is mentioned for ventilation homogeneity but needs an explanation in the legend. Same for ICU, IL6, and Il10 and TNF-alpha. This applies in Appendix 2 and 3

Unfinished sentence – page 11, line 239 ‘Which predicts a continuous problem of research waste in 240 the future [35].’ Same for page 12, line 269 ‘While the European Respiratory Society 270 (ERS) guidelines encouraged researching the effectiveness of physiotherapy in terms of 271 accessibility, patient preference and adherence [10].’

Not sure that I would describe crossover designs as inappropriate for physiotherapy studies as sometimes the way they are conducted and the adequate washout period is fine, particularly if the sample size is difficult to achieve for an RCT design. Consider rephrasing here.

Page 11 – define CONSORT in full

Authors mention the lack of consensus on important outcomes, they could also refer to research priorities which have been published by the ERS and USA bronchiectasis registry, as a way of drawing on what knowledge or other suggestions exist for outcomes as well, as a broad guide.

Authors could consider joining 2 paragraphs on page 13, paragraph 2 and 3 together.

Same for the paragraphs on page 14, the first 2 could be joined together, as the first paragraph is quite short.

Minor points

Page 2, line 37 – add ‘and’ before home exercise program. Line 45 – 18 should be written in full at the start of a sentence.

Page 4 – line 63 – UK should be written in full

Numbers less than 10 should be written in full? If this is the case for this journal, it needs to be attended to throughout the manuscript. There is inconsistency within the manuscript.

Page 4, ine 84 – replace wasn’t, with was not.

A mix of tenses are used in the introduction – would be better to describe the aim of the study was ‘was to identify’, given it has already happened. In the methods – commentary on the Full publication, pilot studies, protocols….. are included – should read as were included. Suggest reviewing the overall manuscript to gain consistency in tense use throughout.

Page 5 – line 101 – studies should be ‘Studies’

Page 6, line 133 – define PRISMA

Page 7, line 156 ‘two reviewers agreed upon a unique name..’ Consider altering the wording for this sentence to improve clarity.

Page 8, line 181, unclear why table (2) in written in this way? Same for figure (4)?

Page 8, line 182 – CCT should be written as (CCT), as it is an abbreviation.

Typo page 13, line 303 – durnig should read during

Reviewer #2: This is an interesting systematic review, with a strong methodology examining outcome reporting in clinical trials of physiotherapy in bronchiectasis. Some suggestions below outline points requiring further clarification.

Abstract, page 2, line 45 – 18 should be written as eighteen at the start of a sentence. Same for line 177 – 74 should be written as Seventy-four…

There is a trend in this manuscript to introduce abbreviations prior to introducing their full name. Suggest reversing this order throughout the manuscript. This is evident in the introduction, with reference to COS-PHyBE, but no explanation of what this is. Same for RCTS later in the methods. Other times this is reversed. Attention for consistency is needed on this point.

Page 4, line 64 – should read ‘5-year mortality rate’ or ‘mortality at 5 years is…’

Page 4, line 84 replace wasn’t with was not

Page 5, line 101- studies should read Studies…

There is some inconsistency in the tense used in writing – should be written ideally in the past tense when describing the methodology throughout.

Page 5 – line 108 – should read ‘studies published only as conference abstracts (plural).

Some correction to grammar is needed in the sentence on page 7, line 140. Also on line 156 – ‘two reviewers agreed upon a unique name for each outcome’. Other aspects of grammar would benefit from further attention within the manuscript with appropriate use of , and ; within sentence. At times, there is mixed use.

Numbers less than 10 are generally written in full in academic writing. It would be worthwhile for the authors to check the guidelines for this journal on this point.

Incomplete sentence on page 9, line 192 – ‘While 23 compared one form of physiotherapy to another…’

It is mentioned that there are 3 categories of airway clearance techniques, but these have not been described in the introduction or earlier in the methodology.

The authors did not appear to include respiratory muscle training as a physiotherapy treatment, but it is not in the list for exclusion – can the authors clarify this point? Or can the authors justify why they included it under the rehabilitation title in Figure 2.

When the authors are specifying lung function as being the most common measure, it would be helpful in the text to know if this is static and dynamic measurements or dynamic measurements predominantly.

There are a number of abbreviations in Appendix 3 – should these be defined in a legend?

Incomplete sentence on page 11, line 239 – ‘Which predicts a continuous problem of research waste in the future [35]’

Page 13, line 303 – typo – durnig should read during

Once an abbreviation is introduced, it is ideal if it can be consistently used – such as ACBT (written out in full again on page 13, line 306, but the abbreviation for this has been introduced earlier.

6. PLOS authors have the option to publish the peer review history of their article (what does this mean?). If published, this will include your full peer review and any attached files.

Reviewer #1: No

Reviewer #2: No

---

## [Author Response · Author response to Decision Letter 0]

23 Dec 2022

Please refer to the attached response to reviewers' file

---

## [Decision Letter · Decision Letter 1]

14 Feb 2023

Evaluation of outcome reporting in clinical trials of physiotherapy in bronchiectasis: the first stage of core outcome set development

PONE-D-22-28058R1

Dear Dr. Hamzeh,

We’re pleased to inform you that your manuscript has been judged scientifically suitable for publication and will be formally accepted for publication once it meets all outstanding technical requirements.

Kind regards,

Brenda M. Morrow, PhD

Academic Editor

PLOS ONE

Additional Editor Comments (optional):

Reviewers' comments:

Reviewer's Responses to Questions

**Comments to the Author**

1. If the authors have adequately addressed your comments raised in a previous round of review and you feel that this manuscript is now acceptable for publication, you may indicate that here to bypass the “Comments to the Author” section, enter your conflict of interest statement in the “Confidential to Editor” section, and submit your "Accept" recommendation.

Reviewer #1: All comments have been addressed

2. Is the manuscript technically sound, and do the data support the conclusions?

Reviewer #1: Yes

3. Has the statistical analysis been performed appropriately and rigorously? 

Reviewer #1: N/A

4. Have the authors made all data underlying the findings in their manuscript fully available?

Reviewer #1: Yes

5. Is the manuscript presented in an intelligible fashion and written in standard English?

Reviewer #1: Yes

6. Review Comments to the Author

Reviewer #1: The authors have addressed all comments. No further feedback is required at this time. Congratulations on addressing these comments thoroughly.

7. PLOS authors have the option to publish the peer review history of their article (what does this mean?). If published, this will include your full peer review and any attached files.

Reviewer #1: No

---

## [Editor Report · Acceptance letter]

7 Mar 2023

PONE-D-22-28058R1 

Evaluation of outcome reporting in clinical trials of physiotherapy in bronchiectasis: The first stage of core outcome set development 

Dear Dr. Hamzeh:

I'm pleased to inform you that your manuscript has been deemed suitable for publication in PLOS ONE. Congratulations! Your manuscript is now with our production department. 

Kind regards, 

on behalf of

Professor Brenda M. Morrow 

Academic Editor

PLOS ONE